# The *Adaptive Host Manipulation* Hypothesis: Parasites Modify the Behaviour, Morphology, and Physiology of Amphibians

Irene Hernandez-Caballero [1], Luz Garcia-Longoria [1], Ivan Gomez-Mestre [2] and Alfonso Marzal [1,3,*]

1   Department of Anatomy, Cellular Biology and Zoology, University of Extremadura, Avenida de Elvas s/n, 06006 Badajoz, Spain
2   Estación Biológica de Doñana, Consejo Superior de Investigaciones Científicas, Avda. Americo Vespucio s/n, 41092 Sevilla, Spain
3   Grupo de Investigaciones en Fauna Silvestre, Universidad Nacional de San Martín, Jr. Maynas 1777, Tarapoto 22021, Peru
*   Correspondence: amarzal@unex.es

**Abstract:** Parasites have evolved different strategies to increase their transmission from one host to another. The *Adaptive Host Manipulation* hypothesis states that parasites induce modifications of host phenotypes that could maximise parasite fitness. There are numerous examples of parasite manipulation across a wide range of host and parasite taxa. However, the number of studies exploring the manipulative effects of parasites on amphibians is still scarce. Herein, we extensively review the current knowledge on phenotypic alterations in amphibians following parasite infection. Outcomes from different studies show that parasites may manipulate amphibian behaviours to favour their transmission among conspecifics or to enhance the predation of infected amphibians by a suitable definite host. In addition, parasites also modify the limb morphology and impair locomotor activity of infected toads, frogs, and salamanders, hence facilitating their ingestion by a final host and completing the parasite life cycle. Additionally, parasites may alter host physiology to enhance pathogen proliferation, survival, and transmission. We examined the intrinsic (hosts traits) and extrinsic (natural and anthropogenic events) factors that may determine the outcome of infection, where human-induced changes of environmental conditions are the most harmful stressors that enhance amphibian exposure and susceptibility to parasites.

**Keywords:** anura; behavioural manipulation; limb malformations; morphological manipulation; parasite transmission; physiological manipulation; urodele



## 1. Introduction

Pathogens frequently parasitise individuals from all vertebrate taxa, imposing negative consequences on the survival and reproductive success of their hosts [1]. Organisms from all groups have adopted this strategy to increase their fitness, including viruses, bacteria, archaea, plants, and animals [2], causing a variety of negative effects on their hosts, from slight detrimental effects to their host (e.g., *Rhabdias* nematodes affect locomotor performance of its hosts [3]) to deadly effects on their hosts (e.g., *Batrachochytrium, salam, rivorans*, (Martel, Blooi, Bossuyt & Pasmans, 2013), a chytrid fungus responsible for massive mortalities in urodele species in Europe [4]).

Parasites and hosts are constantly co-evolving to overcome the respective adaptations achieved [5]. In this co-evolutionary arms race, pathogens and parasites follow different strategies to increase their transmission from one host to another, and hence maximise their fitness. One of these strategies is described in the *host manipulation hypothesis* (also named the *parasite manipulation hypothesis*), which states that parasites may induce alterations of host phenotypic traits (such as behaviour, morphology, or physiology) to increase their survival and transmission success [6,7]. The appearance of extraordinary strategies such as changes in the behaviour of hosts infected by parasites or anti-parasite behaviours

performed by hosts to avoid infections can be considered results of this co-evolutionary arms race [8,9]. The wide range of host–parasite interactions in vertebrates facilitate the exploration of different aspects of the parasite manipulation hypothesis. For example, *Toxoplasma gondii* (Nicolle & Manceaux, 1908), a sexually transmitted parasite, enhances sexual attractiveness of infected brown rat males *Rattus norvegicus* (Berkenhout, 1769) to healthy females [10], thus favouring the transmission of *T. gondii* among conspecifics. Similarly, *Toxoplasma*-infected chimpanzees *Pan troglodytes troglodytes* (Blumenback, 1779) lose their natural aversion towards the urine of leopards *Panthera pardus* (Linnaeus, 1758), thus favouring the trophic transmission of the parasite to this feline species (the final hosts for *T. gondii*) [11].

Parasites can alter hosts' traits to increase the odds of transmission, but not all the phenotypic changes experienced by the host following infection could be considered as a result of adaptive parasite manipulation, as some changes are rather simply side-effects of the infection [12,13]. However, they could be also beneficial for the transmission of the parasite they harbour, and thus distinguishing adaptive physiological changes induced by the parasites from other outcomes of infection could be challenging. For example, it has been proposed that some clinical symptoms of malaria infection (i.e., increased body temperature and $CO_2$ production) may show coincidental benefits for *Plasmodium* transmission, as they favour mosquito host-searching and location [8] and hence enhance parasite transmission [6,14]. In order to identify host phenotypic changes that are truly the result of adaptive manipulation caused by parasite infections, Poulin [15] proposed that such changes should be provoked by the parasite's production and release of molecules that provokes the observed changes in hosts. Importantly, host phenotypic changes should confer increased fitness to the parasite and enhance its transmission.

Host manipulations usually have different dimensions. First, host behaviour can be modified by parasites to increase their transmission and hence increase the parasite fitness. These behavioural changes usually decrease the host fitness by compromising its survival. Parasites transmitted through trophic chains by ingestion are known to manipulate their intermediate host to increase their exposure to predators (definite hosts) [9,16]. For example, *T. gondii* has been suggested to suppress the aversion of its intermediate hosts, the brown rat, to cat odour, thus facilitating the parasite transmission to its felid definite host [17]. Second, parasites may also induce morphological modifications on their host (i.e., colour alteration or creation of new morphological structures) to increase its transmission [6,18,19]. For instance, the tropical arboreal ant *Cephalotes atratus* (Linnaeus, 1758) changes its abdomen colouration to bright red once infected with the nematode *Myrmeconema neotropicum* (Poinar & Yanoviak, 2008). The abdomen of the infected ant resembles the berries of the Bully tree *Hyeronima alchorneoides* (Allemao, 1848) that are usually ingested by an avian definite host [20]. Finally, host physiology may also be modified by parasite infection and increase the pathogen transmission. This physiological imbalance caused by parasites can be the origin of further phenotypic changes in the host. In this sense, it has been suggested that parasites may produce chemicals that interact with the host nervous system and muscles, provoking aberrant behaviours that enhance parasite transmission [21]. For instance, the parasitic Nematomorph hairworm *Spinochordodes tellinii* (Camerano, 1888) produces molecules acting directly on the development of the central nervous system of its host, the terrestrial grasshopper *Meconema thalassinum* (De Geer, 1773). These physiological changes result in abnormal behaviour, making infected insects more likely to jump into an aquatic environment, where the adult parasite reproduces [22]. Multidimensional modifications (modification of more than one phenotypic trait) are common, as morphology, behaviour, and physiology are often interconnected [6] and consequently likely to be jointly affected by parasitic infection [23].

Amphibians are found in a variety of freshwater aquatic and terrestrial environments on every continent except Antarctica. Of the about 8461 described species, 7474 belong to the order Anura (frogs and toads), 773 species are within the order Caudata (salamanders and newts), and only 214 species of caecilians belong to the order Gymnophiona [24]. This

group of vertebrates shows some attributes making them very sensitive to changes in their environment, such as a bi-phasic life cycle with dependence on both aquatic and terrestrial habitat, thin permeable skin and egg membranes that make them vulnerable to pollutants, and an aquatic larval stage increasingly at risk as the hydrological cycle of ponds is shortened [25,26]. For this reason, amphibian species are recognized as important biomarkers for environmental quality [27–29].

Amphibian populations have often experienced severe declines worldwide in the last decades. The International Union for Conservation of Nature (IUCN) estimates that over 40% of amphibian species are threatened with extinction [30], being nowadays the most endangered vertebrate group [31,32]. Together with habitat destruction and chemical pollution, one of the leading causes of this decline is the emergence of new infectious diseases (Emerging Infectious Diseases, *EIDs*) caused by parasites and pathogens (i.e., chytridiomycosis or ranavirus diseases). These parasites negatively affect amphibian populations [33,34] and ultimately impair ecosystem dynamics [35].

Because amphibians inhabit both aquatic and terrestrial habitat, they are exposed to a wide range of parasites in these environments throughout their lifetime (see review in [36]). Moreover, amphibian larvae increase corticosterone (CORT) or thyroid hormone (TH) levels during metamorphosis [37], which may induce temporal immunosuppression and cause amphibian larvae to be more vulnerable to pathogen infection [38]. Amphibians are commonly infected by vector-borne blood parasites, including intraerythrocytic parasites from different genera (i.e., *Heamogregarina*, *Lankesterella*, *Schellackia*, *Aegyptianella*, *Hepatozoon*, *Plasmodium*, and *Haemoproteus*) [39–42] and extraerythrocytic parasites such as species of trypanosomes and microfilariae [43] (see review in [44]). The most common vectors of blood parasites in amphibians are mosquitos, leeches, and ticks [45]. They can also act as paratenic hosts, favouring the transmission of parasites to predators through the consumption of infected amphibians. Macroparasites such as trematodes, nematodes, monogeneans, cestodes, acanthocephalans, and nematomorphs have also been reported to parasitise amphibians, causing detrimental effects that compromise their growth, induce organ damage, and ultimately affect their survival (see review in Ref [36]).

However, despite the diversity of parasites infecting amphibians and the increasing interest in adaptive host manipulation studies, the number of studies exploring the phenotypic changes following infection on this threatened group is still scarce when compared to other vertebrates such as mammals [46] and birds [47]. Here we aim to review the existing literature examining the behavioural, morphological, and physiological changes provoked by parasites in amphibians. We also critically discuss the fitness benefits obtained by parasites from the host phenotypic changes considering the adaptive host manipulation hypothesis.

## 2. Material and Methods

Our literature search was conducted in June 2022. Initial title, keyword, and abstract screening was performed using the items for systematic review and meta-analysis established in PRISMA [48] modified for Ecology and Evolution, PRISMA-Eco Evo [49]. A systematic search on the available literature on parasitism and adaptive manipulation of hosts in amphibians was performed on the *Web of Science* and *Scopus* databases. The search was conducted in English. The original sample for this study included all journal articles published between April 1990 and June 2022 searching with specific keywords and key phrases. The search string comprised three substrings. The first substring targeted morphological effects of parasitism and adaptative manipulation using the following Boolean search keywords [(Amphibian OR anuran OR frog OR urodele OR salamander OR newt) AND parasit* AND (malform* OR deform*)]; we retrieved 345 articles on WoS and 116 articles on Scopus. After curating the search to delete duplicates, we found a subtotal of 368 papers. The second substring aimed at behavioural effects of parasitism and adaptive manipulation using the following Boolean search keywords [(Amphibian OR anuran OR frog OR urodele OR salamander OR newt) AND parasit* AND manipulat* AND behavio*]; we retrieved 72 articles on WoS and 19 articles on Scopus. After the removal of duplicates

between databases, we obtained a subtotal of 72 papers. The third substring aimed at physiological effects of parasitism and adaptive manipulation using the following Boolean search keywords [(Amphibian OR anuran OR frog OR urodele OR salamander OR newt) AND parasit* AND manipulat* AND physiolog*]; we retrieved 117 articles on WoS and 14 articles on Scopus, resulting in a subtotal of 118 papers upon eliminating duplicates. The resulting 558 articles were subsequently filtered using the inclusion/exclusion criteria described in the next section.

*Study Selection Criteria*

We built a decision tree to guide the screening process at the title, keyword, and abstract stage. We followed the guidelines for systematic search and study screening for literature reviews in ecology and evolution proposed by Foo et al. [50] for question formulation, literature search, and screening. The inclusion and exclusion criteria were adapted from scoping searches of previous reviews about parasitism and adaptive manipulation [6,9] (Figure 1).

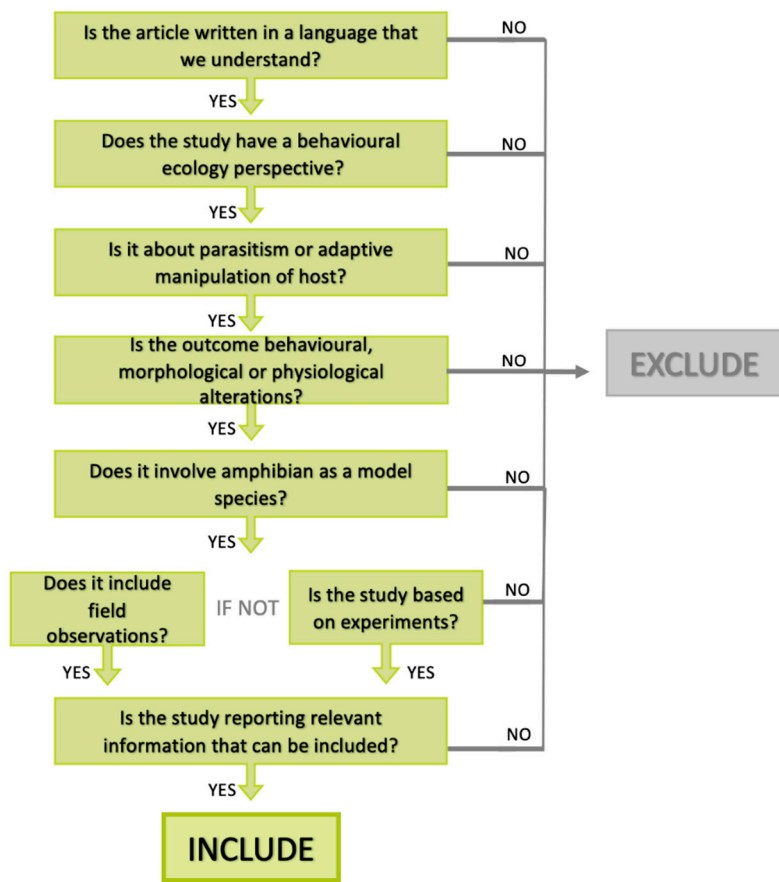

**Figure 1.** PRISMA flow diagram showing inclusion and exclusion criteria adapted for systematic review on parasitism and adaptive manipulation in amphibians.

A total of 62 articles met the inclusion criteria. Finally, these full-text studies showing parasite–amphibian interactions were then screened for their eligibility based on the aforementioned Poulin's basic criteria on adaptive manipulation [15], mainly on how the parasite may enhance its own fitness by subverting host phenotype.

## 3. Results

Twenty-two studies met all our inclusion criteria. All these studies examined the phenotypic manipulation of amphibian hosts following parasite infection. In light of the adaptive manipulation hypothesis, we reviewed the current literature exploring the suite of behavioural,

morphological, and physiological changes induced by parasites on different amphibian hosts to enhance their transmission. We also analysed the intrinsic and extrinsic factors that have been suggested to influence the outcome of parasite infection on amphibians.

*3.1. Behavioural Manipulation*

Some studies have shown that parasites may induce changes in the behaviour of amphibians to favour their transmission and hence obtain fitness benefits (Table 1).

For example, it is known that the trophically transmitted trematode *Ribeiroia ondatrae* (Looss, 1907) can manipulate the behaviour of its amphibian host, the Pacific chorus frog *Pseudacris regilla* (Baird and Girard, 1852), to increase its transmission to its final host. In this line, Goodman and Johnson [51] reported that the trematode infection provoked limb abnormalities in *P. regilla* individuals that could impair the escape behaviour from predators. They showed that infected frogs with malformations waited longer before escaping from simulated predation and allowed the artificial predator approached them to a closer distance, which could facilitate the predation of the infected frogs and hence the transmission of *R. ondatrae* to its final host.

Moreover, they also showed that the trematode infection provoked phenotypic changes (i.e., malformations) that significantly affected how amphibian hosts interact with their surrounding environment, which could have important implications on their survival. For example, they observed that, relative to normal frogs, malformed animals used ground microhabitats more frequently than vertical refugia and perched closer to the ground. These are open and less complex habitats with fewer hiding spots (i.e., bare ground, leaf litter, or small bushes such as juncos or grasses) and with higher predation risk. Hence, the observed changes in habitat use and the reduction in the ability of malformed frogs to utilize such refugia could facilitate their capture by snakes and aerial predators such herons, which is the final host of *R. ondatrae* [51]. Finally, they reported that the infection with *R. ondatrae* also altered the thermoregulation behaviour of *P. regilla* individuals. Infected frogs experienced more difficulties in occupying thermal refugia, and therefore they had higher body temperature. Because temperatures above the critical thermal threshold can compromise locomotor performance and impair an amphibian's ability to escape from predators [52], the altered thermoregulation induced by the parasite can contribute to an increase in the vulnerability of infected *P. regilla* to predators [51].

Similarly, Finnerty et al. [53] explored whether a direct life cycle parasite, the lungworm *Rhabdias pseudosphaerocephala* (Kuzmin, Tkach & Brooks, 2007), could manipulate the thermoregulation and defecation preference behaviour of its amphibian host, the cane toad *Rhinella marina* (Linnaeus, 1758) to increase rates of production and transmission of parasite larvae. They medicated cane toads with an antihelminthic drug to experimentally reduce the infections of lungworms and compared subsequent behaviours of medicated toads versus toads that retained infections in experimental trials. Infected toads selected areas with higher temperatures and spent more time in these warmer environments, therefore increasing their body temperature. Faecal and post-mortem analyses revealed that toads with higher temperatures showed higher lungworm parasitemias and increased lungworm larval production, thus suggesting that parasites may induce anurans to select warmer environments and hence maximise the rates of lungworm egg production. This selection of warmer habitats by infected toads that would increase host temperature could alternatively reflect an adaptive behaviour on the part of the host to kill the parasites (e.g., environmentally induced fever). However, in favour of adaptive manipulation of the host by parasites, laboratory trials have shown that increased toad temperature do not kill or damage the pathogen or minimise its further proliferation and transmission [53]. Moreover, infected toads defecated in moister habitats (closer to water) and produced faeces with higher moisture content, thereby enhancing survival of larvae shed in faeces and increasing the likelihood of encounters with new hosts. These field observations were only registered during the dry season. Heavy rainfall during the rainy season in Australia resulted in a

higher availability of damp refugia, and hence no differences in selection of refugia between infected and uninfected toads were observed.

The metacercariae of the trematode *Codonocephalus urnigerus* (Rudolphi, 1899) are known for inducing alterations of the gonads of its second intermediate host, the marsh frog *Pelophylax ridibundus* (Pallas, 1771) [54,55]. Heavily infected individuals present suppressed sexual and territorial behaviour, do not respond to threats, and show reduced foraging reflexes and motor activity [55]. Therefore, infected frogs do not show anti-parasite behaviour, making them more vulnerable to predators. However, whether these behavioural alterations would increase the chances of infected frogs being captured by bitterns, and hence enhance the transmission of *C. urnigerus* to its final host, requires further investigations.

Chemical signals (i.e., pheromones) are widely used by amphibians, especially salamanders, to exchange information during social interactions, such as territorial combat or mating [56]. Anthony et al. [57] reported that attachment of the chigger mite *Hannemania dunni* (Sambon, 1928) near the snout of salamanders such as the Rich Mountain salamander *Plethodon ouachitae* (Dunn & Heinze, 1933) can damage their nasolabial groove, leading to reduced ability to detect prey and pheromones from conspecifics, such as rivals or potential mates. Moreover, Maksimowich and Mathis [58] explored if high loads of the chigger mite *Hannemania eltoni* (Sambon, 1928) alter the agonistic behaviour of male Ozark zigzag salamanders *Plethodon angusticlavius* (Grobman, 1944) to enhance their aggressiveness during combat. They experimentally analysed whether agonistic behaviour is associated with parasitemia (high or low loads). They measured the time spent in all-trunk raised (ATR) posture of males to assess their level of aggression. Overall, males with lower parasite loads performed a stronger ATR response towards conspecifics than males with higher parasite loads. Males with high parasitic load performed stronger ATR responses only towards highly infected males, but not to lower infected ones. The enhanced aggressiveness of *H. eltoni*-infected males during territorial combat could favour the pathogen transmission through direct contact between hosts.

The anti-parasite behaviour of avoiding territories of highly parasitized conspecific males could be beneficial to females, as ectoparasites can be transmitted during mating [59]. The alteration of anti-parasite behaviour in infected females would favour parasite transmission to non-infected males. In this sense, Maksimowich and Mathis [60] suggested that the chigger mite species *H. eltoni* influences the anti-parasitic behaviour of female Ozark zigzag salamanders towards pheromones of parasitized males. They simultaneously exposed parasitized and non-parasitized males and females to territorial pheromonal markers (i.e., faecal pellets) from infected males with low or high parasite loads, using chemical blanks as controls. They reported that non-infected females spend more time near faecal pellets from infected males with low *H. eltoni* loads. Moreover, these non-infected females seem to avoid highly infected males, as they spend less time near faecal pellets of infected males with high *H. eltoni* loads. On the contrary, infected females spend the same time in control pellets than in pellets from parasitized males (with either low or high parasite loads). Furthermore, they also showed in the same behavioural trials that males of Ozark zigzag salamander with low parasite loads spent more time near control pellets, whereas males with high parasite loads spent more time close to feacal pellets of males with high parasitemias.

Acoustic signalling plays an important role in sexual selection in anurans [56,61]. During breeding season, anuran males produce calls to attract females and reproduce. These calls displayed by males are costly to produce in terms of energy and resources, and thus only males with good quality and enough resources can produce these costly signals. Additionally, calling display could be a handicap since these signals could attract predators [62]. Long and rapid calls produced by male anurans are perceived as honest signals of the quality of individuals, and a larger number of females are attracted towards males producing longer calls with higher call rates [61]. Deuknam and Waldman [63] assessed whether the chytrid fungus *Batrachochytrium dendrobatidis* (Longcore, Pessier & Nichols, 1999) may manipulate the sexual display of male Japanese tree frogs *Hyla japonica* (Günther, 1859) to increase its transmission. They recorded advertisement vocalizations of

male Japanese tree frogs in the field and analysed call properties of males as a function of their infection status. Their outcomes showed that infected males called more rapidly and produced longer calls than their uninfected competitors. During breeding season, male and female frogs go to water bodies where they reproduce, and females release eggs clutches. Since *B. dendrobatidis* zoospores are transmitted through water, the higher investment in calling behaviour in infected frogs could result in the attraction of a larger number of females that consequently become infected by the fungus.

In ecological communities, the complex interactions between parasites, predators, and prey are important drivers of infection and population dynamics in many systems. It has been shown that some parasites may impair the anti-predatory behaviours of their hosts, which could have important consequences for predator–prey interactions. Predators can also acquire pathogens from prey, and hence predators would benefit from recognizing and avoiding consumption of infected prey. Han et al. [64] examined whether the anti-predator behaviour in tadpoles of four frog species was affected by predator chemical cues and *B. dendrobatidis* infection. They showed that infected tadpoles of one frog species *Anaxyrus boreas* (Baird & Girard, 1852) were more active and sought refuge more frequently when exposed to predator chemical cues, thus making captures more difficult to predators. However, infected tadpoles from the other three frog species (*Rana aurora* (Baird & Girard, 1852), *Rana cascadae* (Slater, 1939) and *P. regilla*) did not manifest anti-predator behaviour when exposed to predator chemical cues, which could facilitate predation and hence pathogen transmission. They also explored whether two predator species, the rough-skinned newt *Taricha glutinosa* (Skilton, 1849) and the long-toed salamander *Ambystoma macrodactylum* (Baird, 1950), respond to infection risk by consuming uninfected prey more frequently compared to infected prey. Neither *T. granulosa* nor *A. madrodactylum* avoided consuming *B. dendrobatidis*-infected tadpoles, which could increase their own infection risk.

Coprophagy is a strategy very common among amphibian larvae in which individuals consume conspecific's faeces to obtain nutrients. *Candida humicola* (Diddens & Lodder, 1942) is an intestinal parasitic yeast of anuran tadpoles with a direct life cycle where infective cells are transmitted through the ingestion of faeces by tadpoles. Lefcort and Blaustein [65] determined the thermal preferences of uninfected and infected tadpoles of the red-legged frog (*R. aurora*) in the field. They showed that infected *R. aurora* tadpoles tended to occupy shallow warm-water environments, which can lead to a high concentration of tadpoles and higher accumulation of faeces, which enhances *C. humicola* transmission through coprophagy [66]. They also observed that tadpoles infected with *C. humicola* reduced the anti-predator behaviour of *R. aurora*. Infected tadpoles were not able to discriminate between dangerous and innocuous predator chemical cues and ultimately suffered increased predation by the rough-skinned newt (*T. granulosa*) [65].

Finally, the generalist gram-negative bacteria *Aeromonas hydrophila* (Chester, 1901) is responsible for red-leg syndrome in amphibians [67]. These bacteria may induce changes in the anti-predator behaviour of bullfrogs *Rana catesbeiana* (Shaw, 1802). Lefcort and Eiger [68] experimentally infected bullfrog tadpoles with alcohol-killed *A. hydrophila* to test whether the induced response to infection may alter the ability of tadpoles to detect and avoid capture by predatory salamanders (*T. granulosa*). They showed that infected bullfrog tadpoles performed reduced refuge-seeking behaviour in presence of predator *T. granulosa*. This altered anti-predatory behaviour can lead to increased predation that enhances bacteria transmission to suitable amphibian hosts.

**Table 1.** Summary of studies reporting behavioural manipulations in several amphibian hosts induced by parasites from different taxa.

| Parasite Species (Taxonomic Class) | Host Species | Parasite Life Cycle | Outcome of Host–Parasite Interaction Increasing Parasite Transmission | Effect on Infected Host | References |
|---|---|---|---|---|---|
| *Ribeiroia ondatrae* (Trematoda) | *Pseudacris regilla* | Indirect | Enhanced predation risk of infected host | Loss of fear to predator Microhabitat altered choice Altered thermoregulatory behaviour | [51] |
| *Codonocephalus urnigerus* (Trematoda) | *Pelophylax ridibundus* | Indirect | Enhanced predation risk of infected host | Altered sexual, territorial, and foraging behaviour Failed anti-predator behaviour | [55] |
| *Rhabdias pseudosphaerocephala* (Secernentea) | *Rhinella marina* | Direct | Toads with higher temperatures showed higher lungworm parasitemias and increased lungworm larval production Increased parasite larvae survival in moist environments | Selection for warmer environments Preference for moist habitat for defecation | [53] |
| *Hannemania eltoni* (Arachnida) | *Plethodon angusticlavius* | Indirect | Increased contact between host and parasite | Increased agonistic behaviour in males Failed anti-parasitic behaviour in females | [58,60] |
| *Hannemania dunni* (Arachnida) | *Plethodon ouachitae* | Indirect | Increased contact between host and parasite | Reduced ability to detect prey and pheromones from conspecifics | [57] |
| *Batrachochytrium dendrobatidis* (Chytridiomycetes) | *Hyla japonica* *Anaxyrus boreas* *Taricha glutinosa* *Ambystoma macrodactylyum* | Direct | Increased contact between infected host during mating Enhanced predation risk of infected host | Enhanced calling behaviour in males Erratic swimming behaviour Failed detection of predator cues | [63,64] |
| *Candida humicola* (Saccharomycetes) | *Rana aurora* | Direct | Increased parasite transmission through coprophagy Enhanced predation risk of infected host | Preference for environments with high concentration of conspecifics and higher accumulation of faeces Failed detection of predator chemical cues | [65] |
| *Aeromonas hydrophila* (Gammaproteobacteria) | *Rana catesbaiana* | Direct | Enhanced predation risk of infected host | Reduced refuge-seeking behaviour in presence of predator | [68] |

*3.2. Morphological Manipulation*

Morphological manipulations in amphibians due to parasites were first reported by Sessions et al. [69]. Since then, experimental infections and field studies have corroborated their findings [51,70]. Alteration of amphibian limb morphology caused by parasites usually leads to impaired locomotor performance, which enhances the transmission of parasites that are mainly transmitted via food chain (Table 2).

For example, the trematode *R. ondatrae* is the most famous example of a parasite that induces morphological changes in the body plan of several species of frogs, toads, and salamanders (Figure 2 and Table 2) [71,72]. Metacercariae of this trematode encyst in the muscle surrounding the area of limb buds of larvae, disturbing the spatial organisation of cells that produce signalling molecules involved in the formation of primary limb axes [73], and triggering cell intercalation during early developmental stages [70]. As a result, infected amphibians show different types of limb malformations when amphibian larvae have fully developed. These deformities caused by *R. ondatrae* lead to poor locomotor performance and reduced survival due to higher predation risk, which can increase the parasite transmission to predators (Figure 3) [51].

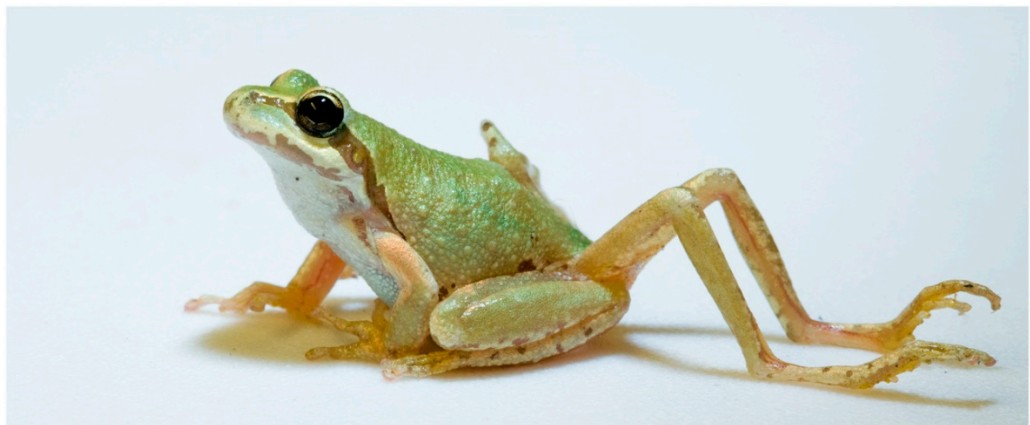

**Figure 2.** Pacific chorus frog (*Pseudacris regilla*) with hind limbs malformation induced by the infection with the trematode *Ribeiroia ondatrae*. Source: Goodman and Johnson, 2011 [51], licensed under CC BY 2.5.

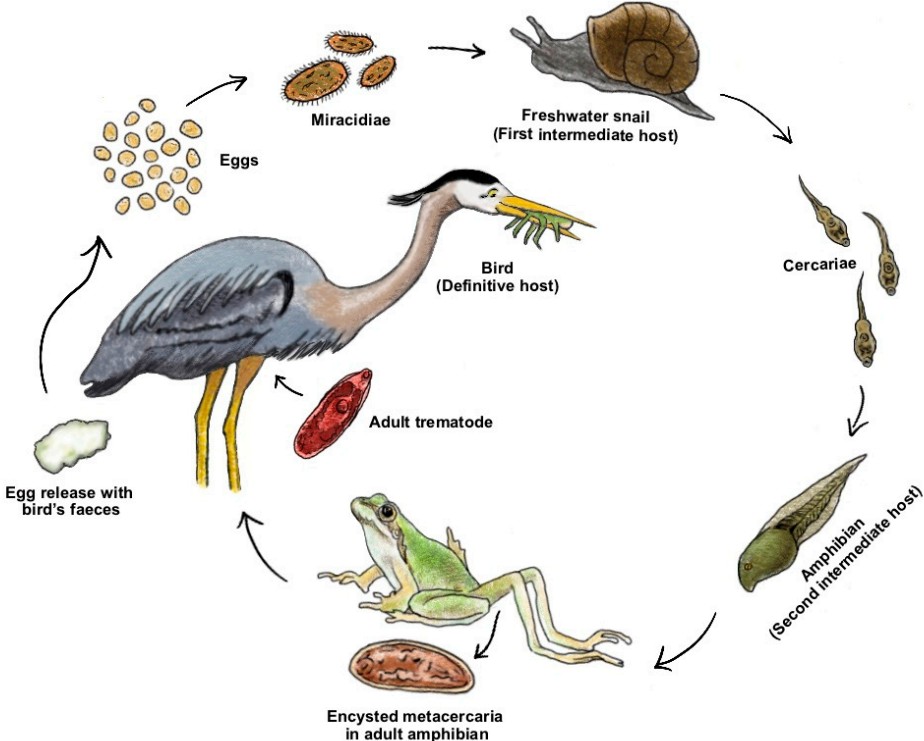

**Figure 3.** The life cycle of the trematode *Ribeiroia ondatrae*. Depicted hosts are as follows: the eggs of the worms are released to the water through the bird's faeces and then hatch in a free-living form (Miracidiae) that infects its first intermediate host, the planorbid snail (*Planorbella* sp.). After a series of stages inside the aquatic mollusk, another swimming form (Cercariae) is shed and infects the second intermediate host (Ranid tadpoles). Cercariae form cysts that impair the normal limb development in adult amphibians, causing deformities. Adult frogs with malformed limbs are more vulnerable to predation by the final host, the great blue heron *Ardea Herodias* (Linnaeus, 1758). Source: Irene Hernandez-Caballero, 2022.

Recently, Svinin et al. [74] analysed the effect of trematode *Strigea robusta* (Szidat, 1928) infection in the gonadal structure and limb development in tadpoles of three toad species: *Bufo bufo* (Linnaeus, 1758), *Bufotes viridis* (Laurenti, 1768), and *Bufotes baturae* (Stöck, Schmid, Steinlein & Grosse, 1999). They conducted an experiment with nine groups containing an equal number of tadpoles. Tadpoles from experimental groups were exposed to a fixed number of cercariae, whereas tadpoles from control groups were not exposed to

parasites. To test for gonadal morphological differences between infected and uninfected tadpoles, they examined the gonads with a laser scanning confocal microscope. They found that malformations on the hindlimbs (51.8%) were more frequent than on the forelimbs (20.2%). Their outcomes also showed that cercariae induced malformations in 57.9% of tadpoles. Of these tadpoles, 51.8% showed mild cases of limb abnormalities (the limbs were not dramatically affected, and individuals could move normally), whereas 6.1% of tadpoles showed severe malformations on their limbs, impairing normal mobility. The malformations induced by *S. robusta*, especially in the hindlimbs, would interfere with the capacity of tadpoles to successfully escape from predators and may favour its transmission to anatids, with its final hosts of this trematode parasite [75,76]. However, they did not find any differences in stages of gonadal development or heterochromatin distribution within gonocytes between infected and uninfected toad tadpoles. The absence of detrimental effect of *S. robusta* on reproductive structures of the host could be explained, as any abnormal gonadal development of tadpoles can lead to host population decline, which could be detrimental for parasite survival and transmission in toad populations [74].

The specific types of limb abnormalities associated with *Riberoia* infection and their relative frequencies differed substantially among amphibian groups. Johnson et al. [71] showed that 94% of the abnormalities observed in anurans affected the hind limbs. In urodeles, 41.9% of the abnormalities involved the forelimbs and 56.9% involved the hind limbs, likely because in caudata, forelimbs develop externally, not within the body cavity, as in anurans. Experimental infection with *R. ondatrae* cercariae to frogs *Lithobates sylvaticus* (LeConte, 1825) and *R. pipiens* showed that this trematode cercaria preferably seek the limbs of anuran hosts, such as folds near the limb buds and tail [70]. On the contrary, *R. ondatrae* encysts in both forelimbs and hindlimbs in urodeles [77]. These variations in the location of limb malformation among amphibian groups may be explained by the parasite manipulation hypothesis, because these differential abnormalities in anurans and urodeles may differentially increase the likelihood of *Ribeiroia*-infected amphibians to be predated by their own suitable final hosts [77,78]. In this sense, urodeles use all four limbs during locomotion, and metacercariae and malformations were found in all limb groups at similar frequencies, hence debilitating the locomotor activity of these amphibians. On the contrary, frogs use hind legs for leaping and to propel themselves through the water while swimming, and most malformations and metacercariae in anurans were found around the hind limbs, hence impairing their locomotor activity [77,78].

Metacercaria from trematodes other than *R. ondatrae* have been proposed to cause different body abnormalities in their amphibian hosts. For example, *Clinostomum* spp. (Leidy, 1856) is a trematode with an indirect life cycle (similar to *R. ondatrae*), which is trophically transmitted to their final hosts, the members of the family Ardeidae (i.e., herons and egrets) [79]. *Clinostomum* spp. has been reported to induce scoliosis in the tiger salamander *Ambystoma tigrinum* (Green, 1825) [80]. A massive number of *Clinostomum* spp. metacercariae encysted within the skeletal musculature, coelomic cavity, and subcutaneous space were the cause of scoliosis. However, it remains unknown whether this malformation in tiger salamanders increases pathogen transmission or reduces fitness.

Similarly, larval stages of another trematode transmitted via food chain *Acanthostomum burminis* (Bhalerao, 1926) also induce axis deformities under experimental conditions, which may increase the likelihood of being consumed by predators. Some fish and reptile species, such as the freshwater snakes *Xenochrophis piscator* (Schneider, 1799), are final hosts of this parasite. Infected amphibians, which show impaired locomotion, are more likely ingested by predators, hence completing the *A. burminis* life cycle. Also, infected tadpoles of the common hourglass treefrog *Polypedates cruciger* (Blyth, 1852) [81,82] and Asian common toad *Duttaphrynus melanostictus* (Schneider, 1799) [83] showed malformations in the vertebral column such as scoliosis (lateral deviation in the normally straight line of the spine), kyphosis (abnormal convexing of the spine), and extension of the spine beyond the rump. Limb malformations caused by this parasite on *P. cruciger* have also been observed [81], but axis malformations are more frequent. Morphological malformations

induced by this parasite can enhance its transmission to a new host because they decrease the locomotor performance of tadpoles, and therefore facilitate its predation.

**Table 2.** Summary of studies reporting morphological manipulations in several amphibian hosts induced by parasites from different taxa.

| Parasite Species | Host Species | Parasite Life Cycle | Outcome of Host–Parasite Interaction Increasing Parasite Transmission | Effect on Infected Host | References |
|---|---|---|---|---|---|
| *Ribeiroia ondatrae* (Trematoda) | *Taricha torosa* *Tarica granulosa* *Ambystoma macrodactylum* *Bufo boreas* *Pseudacris regilla* *Rana aurora* *Rana luteiventris* *Rana catesbaiana* *Rana cascadae* | Indirect | Increased predation risk of infected host by impaired locomotor activity | Missing limbs and digits, extra limbs, extra appendices, skin webbings fusion | [51,69–72,77,78] |
| *Clinostomum* **spp.** (Trematoda) | *Ambystoma tigrinum* | Indirect | Increased predation risk of infected host by impaired locomotor activity | Scoliosis | [79] |
| *Acanthostomum burminis* (Trematoda) | *Polypedates cruciger* *Duttaphrynus melanostictus* | Indirect | Increased predation risk of infected host by impaired locomotor activity | Extension of the spine Scoliosis Kyphosis | [80,82] |
| *Strigea robusta* (Trematoda) | *Pelophylax ridibundus* *Bufo bufo* *Bufotes viridis* *Bufotes baturae* | Indirect | Increased predation risk of infected host by impaired locomotor activity | Mild and severe limb malformations | [74,76] |

Other freshwater invertebrates, such as copepods, are suspected to induce malformation in amphibians. Following this idea, Kupferberg et al. [84] reported a highly significant association between limb abnormalities in the Foothill Yellow-legged Frogs *Rana boylii* (Baird, 1854) and parasitic copepod *Lernaea cyprinacea* (Linnaeus, 1758) infestation in 2006, but they failed to prove the same effect with the data collected on a survey two years later. However, whether these limb abnormalities lead to enhance copepod transmission remains unknown. The differences in malformation appearance between years in this study indicate the necessity of conducting long-term investigations to exclude stochastic scenarios.

*3.3. Physiological Manipulation*

Parasites may also impose negative effects on the physiology of their hosts. Several investigations in amphibians have revealed that some changes in host physiology associated with parasite infection may enhance pathogen proliferation and transmission, suggesting parasite manipulation (Table 3).

For example, Finnerty et al. [53] conducted an experiment to analyse the effect of the lungworm *R. pseudosphaerocephala* on water content on the faeces of its host, the cane toad. With this aim, they individually housed wild-caught toads and medicated them to clear lungworm infection. They subsequently collected faeces of infected and non-infected (de-wormed) toads to analyse parasite larvae survival in relation to water content on the faeces. They showed that infected toads defecated faeces with higher water content related to non-infected toads. Consistent with predictions from host manipulation hypothesis, they found a higher fecundity and larval survival of lungworms in moister faeces. Thus, these outcomes suggest that the infection with *R. pseudosphaerocephala* alters hydric content of its hosts faeces, hence increasing the parasite reproductive success and fitness.

Leeches are also capable of inducing physiological alterations to avoid host detection while they are feeding. Leeches first attach to their host using their anterior and posterior suckers, and then they use specialized teeth to pierce through the host skin. Once leeches are attached, they break host blood vessels and consume their blood. This process is very invasive, so leech saliva is formed by different compounds secreted to prevent blood coagulation, reduce pain sensation, and downregulate immunocyte activity of their hosts [85,86].

For instance, Durant et al. [87] suggested that the leech species *Placobdella* spp. (Blanchard, 1893) releases chemical signals to disrupt the endocrine system of the Eastern hellbender salamander *Cryptobranchus alleganiensis* (Daudin, 1803) and avoid detection. To test their hypothesis, they conducted an experiment with wild-caught hellbender salamanders. They inoculated individuals with saline or adrenocorticotropin hormone (ACTH), a pituitary hormone that stimulates the inter-renal glands to excrete corticosterone (CORT). They subsequently monitored their plasma corticosterone and white blood cell count for 50 h and compared hormone and cellular changes in relation to leech infection. They found that infection with leeches dampened the CORT secretion response of salamanders to physical restraint, but inoculation with ACTH restored inter-renal CORT responses of hellbenders. Moreover, hellbenders infected with leeches also showed an altered leukocyte count (higher percentage of eosinophils) relative to uninfected hellbenders. They suggested that leeches produce neurotransmitters used for down-regulation of corticosterone release at the level of the pituitary or hypothalamus. Because a reduced CORT response to stressful situations (i.e., parasitism) can minimise host ability to restore homeostasis [88], leeches that successfully manipulate endocrine function of the salamander host would avoid detection, and hence they could feed for a longer period of time. As a result, these leeches would have more access to resources used for growth and future reproduction.

**Table 3.** Summary of studies reporting physiological manipulations in amphibian hosts induced by parasites from different taxa.

| Parasite Species | Host Species | Parasite Life Cycle | Outcome of Host–Parasite Interaction Increasing Parasite Transmission | Effect on Infected Host | References |
|---|---|---|---|---|---|
| *Rhabdias pseudosphaerocephala* (Secernentea) | *Rhinella marina* | Direct | Higher fecundity and larval survival of lungworms in moister feces | Defecated feaces with increased water content | [53] |
| *Placobdella* spp. (Clitellata) | *Cryptobranchus alleganiensis* | Direct | Avoid parasite detection by host; hence, parasites can feed for a longer time | Reduced corticosterone response of hosts to stressful situations | [87] |

### 3.4. Factors Influencing Amphibian–Parasite Interactions

Amphibians and their parasite communities are part of complex ecosystem dynamics that can be altered due to biotic and abiotic factors. In this sense, several intrinsic and extrinsic factors have been suggested to influence the outcome of parasite infection in their amphibian host [9]. Intrinsic factors are individual traits that make hosts more vulnerable to infection, such as alterations of life-cycles, behaviour, age at sexual maturity, and gut microbiota diversity [89]. Extrinsic factors are those that affect the vulnerability to infection, such asp natural events (i.e., seasonal nutrient cycles, ecological interactions) or human-induced alterations (i.e., contamination, habitat fragmentation). Next, we will provide examples of both intrinsic and extrinsic factors modulating parasite infections in amphibians.

### 3.4.1. Intrinsic Amphibian Traits That Influence Infection

Although parasites have evolved mechanisms to facilitate the infection on their hosts, not all amphibians exposed to parasites become infected. Exposition (number of host encounters with parasite infective stages) and susceptibility (proportion of parasites successfully infecting the hosts when exposed to them) have been proposed as important determinants of parasite infection success [89,90]. Exposure time and susceptibility to parasites in amphibians is specially related to the duration of larval period. Amphibian species with longer larval periods have greater exposure to parasites [91], which could lead to higher parasite loads. Nonetheless, amphibian species with longer larval periods may also invest more time and resources to develop higher resistance to infection and clear parasites [92], which could result in fewer parasites at metamorphosis and experiencing less severe pathologies [91]. High parasite loads can lead individuals to assume physiological

trade-offs on the allocation of limited resources [38] during metamorphosis. Individuals can attempt to clear infection by re-allocating limited resources to produce a greater immune response. As a result, these individuals experiment reduction in size and developmental rate. For example, infected tadpoles of *Lithobates pipiens* (Schreber, 1782) experienced reduced growth and development when infected by the trematode *Echinostoma trivolvis* (Cort, 1914) [93]. Another example of ectotherm hosts attempting to clear infections can be found in behavioural fever. Infected host behaviour increases their internal temperature, leading to enhanced host immune response or reduced pathogenic burden [94]. The green tree frog *Litoria chloris* (Boulenger, 1892) has been experimentally reported to clear *B. dendrobatidis* infection by the elevation of body temperatures [95].

Alternatively, amphibian hosts can adjust susceptibility to parasites by performing anti-parasitic behaviour. These are actions used by potential hosts to prevent acquiring infection and the establishment of parasites. For example, tadpoles of the Pacific chorus frog (*P. regilla*) have been reported to experiment bursts of activity to avoid trematode infection. This anti-parasite behaviour displayed by tadpoles includes increased activity, fast swimming, body twisting, and aggressive turning that leads to successfully avoiding infection of echinostomatoid cercariae [96]. Moreover, Daly and Johnson [97] experimentally inhibited the anti-parasite behaviour of Pacific chorus frogs and exposed them to pathogenic trematodes (*Ribeiroia* and *Echinostoma*). They reported that behaviourally impaired tadpoles (i.e., anesthetized) were more likely to become infected and suffer from higher parasite loads once infected.

Pre-existing general differences in animal behaviour can influence individual susceptibility to infection. Amphibians show different behavioural syndromes (correlation among behavioural traits across situations) and personality traits (consistent differences in individual behaviour) that influence infection likelihood and manipulation occurrence. As an example, Wood frog tadpoles (*L. sylvaticus*) showed different personalities towards trematode cercaria infection under experimental conditions [98]. Their personalities influence the infection intensity, where bold and more active individuals had lower parasite loads. This behaviour plasticity can increase host survival, which leads to an increment in host fitness [99].

Finally, some studies have suggested that changes in early-life microbiota of amphibian hosts affect parasite infection risk throughout their life. For example, Knutie et al. [100] reported that experimental manipulation of the gut and skin bacterial communities of the Cuban tree frog (*Osteopilus septentrionalis*) tadpoles affected later-life resistance to the parasitic gut nematode (*Aplectana hamatospicula*) in adult frogs. They conducted an experiment in which juvenile frogs were randomly assigned to one of four water treatments: (i) pond water, (ii) sterile (i.e., autoclaved) pond water only, (iii) sterile pond water and short-term antibiotics, and (iv) sterile pond water and long-term antibiotics. A group of 10 individuals from each water treatment were euthanized to determine their bacterial community. Finally, the remaining juveniles of each group were allowed to develop into adult frogs and then were exposed to *A. hamatospicula*. Their outcomes showed that adult frogs that had reduced bacterial diversity as tadpoles (i.e., treated with antibiotics) had three times more gut nematodes than adult frogs that conserved their microbiota intact as tadpoles. Moreover, they also reported that adult frog microbiota at the time of infection did not affect host resistance to nematodes. Hence, susceptibility to *A. hamatospicula* infection in *O. septentrionalis* is related to early-life alteration of microbiota and not adulthood.

### 3.4.2. Extrinsic Factors That Influence Infection

Extrinsic factors have a high influence on the level of detrimental effects produced by parasitism in amphibians. Natural and human-induced changes in water parameters may increase exposition and susceptibility from host to parasites.

Alteration of aquatic ecosystems may induce changes in amphibian exposition to parasites. For example, nitrogen or organic pesticide levels may affect the abundance of intermediate hosts. Larger populations of intermediate hosts may promote the production

of cercariae stages of trematodes. This parasite aggregation enhances the probabilities of amphibian hosts to become infected [101]. For instance, an imbalance in litter nitrogen nutrients may lead to higher eutrophication rates, which have been linked to drastic demographic increments of snails. The upraised numbers of snails produce a higher number of *R. ondatrae* cercariae, thus increasing the probability of infection in amphibians [102].

Amphibian susceptibility to parasites is known to be linked to agricultural runoff. Field and laboratory experiments have shown that pesticide (i.e., Atrazine) exposure induced immunosuppression in *L. sylvaticus* tadpoles. Additionally, this immunosuppression facilitated the infection by *R. ondatrae* cercaria and increased the number of limb malformations [103]. Koprivnikar, [104] suggested that when Atrazine is combined with other ecological stressors (i.e., predation pressure) the effects are particularly negative for parasitized tadpoles. This synergy compromises tadpole survival and has a negative impact on their fitness.

Salt (NaCl) used for road de-icing often leaches into aquatic ecosystems after the snow melts. It has been suggested that NaCl can damage freshwater organisms and induce changes in the outcome of species interactions (i.e., host–parasite interactions). High water salinity is known to increase amphibian susceptibility to parasite infection. Following this line, Buss and Hua [105] tested the susceptibility of *L. sylvaticus* to trematode infection under four different NaCl concentrations (0, 1, 2, and 3 g/L NaCl). They concluded that relative to tadpoles not exposed to NaCl (0 g/L NaCl treatment), exposure to 1, 2, and 3 g/L of NaCl increased parasite infection. Tadpoles exposed to the higher concentration (3 g/L NaCl treatment) had more parasites in comparison to tadpoles exposed to the other treatments with lower NaCl concentrations. They concluded that Wood frog tadpole susceptibility to trematode infection and trematode load were related to NaCl concentration in the water. Similarly, Milotic et al. [106] experimentally showed that *L. sylvaticus* and *L. pipiens* tadpoles exposed to high NaCl concentration presented higher trematode loads and showed reduced anti-parasite behaviour (i.e., reduced activity level in the presence of cercariae) in comparison to tadpoles exposed to lower NaCl concentration.

## 4. Conclusions

Parasites and hosts are constantly co-evolving in continuous arms races that result in the evolution of extraordinary strategies. Thus, parasites may manipulate the host phenotype to enhance pathogen transmission, whereas hosts may exhibit different anti-parasite behaviours to avoid infections. Amphibians are found in very diverse habitats and are exposed to a wide array of parasites and pathogens. During their lifetime, many amphibians experience drastic phenotypic changes, especially metamorphosis, during which they are particularly susceptible to parasite infections. In consequence, similar to other vertebrates, amphibians (especially amphibian larvae), might be the target of parasite infections, and are therefore subject to parasite manipulations.

The determination of the number of parasites and pathogens that might potentially manipulate amphibian behaviour is a major issue in evolutionary biology. In this sense, there are several parasites and pathogens that infect amphibians and follow a manipulative strategy. In our study, we reviewed the current literature providing examples on the *Adaptive Manipulation Hypothesis* in amphibians. We showed that in order to obtain fitness benefits, parasites may induce changes in the behaviour, morphology, and physiology of their amphibian hosts. Although morphological changes due to parasite infection such as limb malformations have been further studied, changes in the behaviour of amphibian hosts following parasite infection are frequent. Moreover, physiological alteration may be involved in the other types of host manipulations since it is the basis of many alterations of behaviour and morphology.

There are several intrinsic factors (hosts traits) and extrinsic factors (natural events and human-induced alterations) that determine the outcome of infection, where anthropogenic changes of environmental conditions are the most harmful, as they act as stressors that

enhance amphibian exposure and susceptibility to parasites. Further studies should focus on the synergies between these factors and their role on parasite manipulation of host.

The lack of information regarding parasitism on less well-known amphibian groups such as caecilians highlights the need for future studies. In this sense, further research should be conducted for a better understanding of the effects of parasite–amphibian interactions and to control the decline of amphibian populations.

**Author Contributions:** Conceptualization, I.H.-C., L.G.-L., I.G.-M. and A.M.; methodology, I.H.-C. and A.M.; writing—original draft preparation, I.H.-C. and A.M.; writing—review and editing, I.H.-C., L.G.-L., I.G.-M. and A.M.; funding acquisition, L.G.-L. and A.M. All authors have read and agreed to the published version of the manuscript.

**Funding:** This study was funded by the Consejería de Economía e Infraestructura of the Junta de Extremadura and the European Regional Development Fund, a Way to Make Europe (research project IB20089).

**Institutional Review Board Statement:** Not applicable.

**Acknowledgments:** We would like to thank Jorge S. Gutiérrez, Casimiro Corbacho and M. Auxiliadora Villegas for their advice during the writing and editing process. We also thank to reviewers and Academic editors for their comments to improve the manuscript.

**Conflicts of Interest:** The authors declare no conflict of interest.

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
