# Peer review of "The Adaptive Host Manipulation Hypothesis: Parasites Modify the Behaviour, Morphology, and Physiology of Amphibians"

_diversity, doi:10.3390/d14090739_

Round 1
Reviewer 1 Report
The ms represents a review on a specific aspect of parasite-amphibian host interactions, the adaptive manipulation hypothesis. I was surprised to see how little is known about this issue. Of course, there are many papers on parasite-anuran host relationships, but the vast majority is descriptive or addressing taxonomic issues with the parasites. Therefore, I believe that this review could stimulated the research area. Besides some minor comments and recommendations written directly into the attached ms I did not find anything to critize. A well-done and well-organized ms, congratulations.
Ulrich SInsch

Author Response
RESPONSE TO REVIEWER 1
Thank you very much.
We have taken into account all the suggested changes written directly into the attached ms by the Reviewer. You can find the suggested changes in the revised version of the manuscript.
Sincerely yours,
Alfonso Marzal
Reviewer 2 Report
The manuscript by Hernandez-Caballero with co-authors is a well-written overview and made according to all rules of MDPI review articles. The article is interesting and is devoted to relevant theme – changes in behavior, morphology and physiology of amphibian hosts under the influence of parasites. I enjoyed reading the article. The materials in the article is presented logically and consistently. The article will be of interest not only to parasitologists, but also to everyone interested in the impact of parasites on the host organism.
It should be noted that the co-evolution of parasites (usually helminths) and their hosts is a rather lengthy process. And it follows the path of smoothing out the antagonism between the parasite and the host. A parasite, especially a specific parasite, is not interested in harming its host, much less causing its death. After all, the parasite needs to grow in the hosts and give offspring. In natural ecosystems, there is no harm from specific parasites for hosts. Only if the parasite needs to change the host to continue its life cycle, then it can negatively affect the hosts: change the behavior of the host or even cause his death. As a rule, these are larval forms of helminths (Codonocephalus spp., Echinostoma spp., Strigea spp.).
I am of an opinion that the manuscript could be published after some minor corrections, which will only make the article better.
1. Lines 21, 55 and others – I would rather use the term “final” than “definitive”. But this is at the discretion of the authors.
2. I propose to slightly change the names of tables 1, 2, 3: Behavioural (or Morphological or Physiological) manipulations in several amphibian hosts induced by parasites from different taxa.
3. Future readers of the article will not only be narrow specialists. Therefore, in these tables (1, 2, 3) it is necessary to add the names of higher systematic taxa (at least) of the given parasitic organisms. For example, for Batrachochytrium salamandrivorans this is Chytridiomycetes or Fungi.
4. According International Code of Zoological Nomenclature (ICZN) at the first mention of animal species (Rattus norvegicus (Berkenhout, 1769), Cephalotes atratus (Linnaeus, 1758) Panthera pardus Linnaeus, 1758 and others) and their parasites (Toxoplasma gondii (Nicolle & Manceaux, 1908), Myrmeconema neotropicum Poinar & Yanoviak, 2008) in the text of article, its full Latin name with the author and year of description should be given. Further, at the second mention of the Latin name of the parasite, the generic name of the parasite is reduced to one letter, the author and the year are not given.
5. Line 376 – I would rather use the term “trematode” than “flatworm”.
6. For example - Line 539 - Cuban tree frog, but better O. septentrionalis.
7. Line 94 - extra point.
In conclusion, I express my opinion. The manuscript can be published after minor corrections.

Author Response
RESPONSES TO REVIEWER 2.
Thank you very much for your review and your nice words. Below we explain how we have responded to the requested changes. Please find enclosed a revised version with the requested changes included. We highlight in bold our responses to facilitate the reviewing process.
1. Lines 21, 55 and others – I would rather use the term “final” than “definitive”. But this is at the discretion of the authors.
R1. Thank you very much. We have replaced "definitive" by "final" through the manuscript.
2. I propose to slightly change the names of tables 1, 2, 3: Behavioural (or Morphological or Physiological) manipulations in several amphibian hosts induced by parasites from different taxa.
R2. Done, thank you.
3. Future readers of the article will not only be narrow specialists. Therefore, in these tables (1, 2, 3) it is necessary to add the names of higher systematic taxa (at least) of the given parasitic organisms. For example, for Batrachochytrium salamandrivorans this is Chytridiomycetes or Fungi.
R3. Thank you. We have added the taxonomic class to all parasitic organisms in Tables 1-3, as suggested.
4. According International Code of Zoological Nomenclature (ICZN) at the first mention of animal species (Rattus norvegicus (Berkenhout, 1769), Cephalotes atratus (Linnaeus, 1758) Panthera pardus Linnaeus, 1758 and others) and their parasites (Toxoplasma gondii (Nicolle & Manceaux, 1908), Myrmeconema neotropicum Poinar & Yanoviak, 2008) in the text of article, its full Latin name with the author and year of description should be given. Further, at the second mention of the Latin name of the parasite, the generic name of the parasite is reduced to one letter, the author and the year are not given.
R4. We have reviewed the manuscript and mention species according to ICZN, as suggested. Thank you.
5. Line 376 – I would rather use the term “trematode” than “flatworm”.
R5. We have replaced "flatworm" by "trematode". Thank you.
6. For example - Line 539 - Cuban tree frog, but better O. septentrionalis.
R6. Done, thank you.
7. Line 94 - extra point.
R7. Thank you. We have deleted this extra point.